# Machine-Learning-Enabled Diagnostics with Improved Visualization of Disease Lesions in Chest X-ray Images

**DOI:** 10.3390/diagnostics14161699

**Published:** 2024-08-06

**Authors:** Md Fashiar Rahman, Tzu-Liang (Bill) Tseng, Michael Pokojovy, Peter McCaffrey, Eric Walser, Scott Moen, Alex Vo, Johnny C. Ho

**Affiliations:** 1Department of Industrial, Manufacturing and Systems Engineering, The University of Texas, El Paso, TX 79968, USA; 2Department of Mathematics and Statistics, Old Dominion University, Norfolk, VA 23529, USA; mpokojovy@odu.edu; 3Department of Radiology, The University of Texas Medical Branch, Galveston, TX 77550, USA; pemccaff@utmb.edu (P.M.); emwalser@utmb.edu (E.W.); stmoen@utmb.edu (S.M.); ahvo@utmb.edu (A.V.); 4Department of Management and Marketing, Turner College of Business, Columbus State University, Columbus, GA 31907, USA; ho_johnny@columbusstate.edu

**Keywords:** convolutional neural networks (CNNs), class activation map (CAM), bacterial/viral infection, disease diagnosis, medical imaging

## Abstract

The class activation map (CAM) represents the neural-network-derived region of interest, which can help clarify the mechanism of the convolutional neural network’s determination of any class of interest. In medical imaging, it can help medical practitioners diagnose diseases like COVID-19 or pneumonia by highlighting the suspicious regions in Computational Tomography (CT) or chest X-ray (CXR) film. Many contemporary deep learning techniques only focus on COVID-19 classification tasks using CXRs, while few attempt to make it explainable with a saliency map. To fill this research gap, we first propose a VGG-16-architecture-based deep learning approach in combination with image enhancement, segmentation-based region of interest (ROI) cropping, and data augmentation steps to enhance classification accuracy. Later, a multi-layer Gradient CAM (ML-Grad-CAM) algorithm is integrated to generate a class-specific saliency map for improved visualization in CXR images. We also define and calculate a Severity Assessment Index (SAI) from the saliency map to quantitatively measure infection severity. The trained model achieved an accuracy score of 96.44% for the three-class CXR classification task, i.e., COVID-19, pneumonia, and normal (healthy patients), outperforming many existing techniques in the literature. The saliency maps generated from the proposed ML-GRAD-CAM algorithm are compared with the original Gran-CAM algorithm.

## 1. Introduction

Lung infections caused by bacteria, viruses, and fungi commonly produce diseases such as pneumonia, tuberculosis, asthma, and others that internationally account for 1.5 million deaths annually. Recently, the world has experienced one of the most transmissive diseases, COVID-19, caused by the SARS-CoV-2 virus, which infects the lungs and mutates rapidly as an RNA virus [1]. This characteristic has created many variants, such as Alpha, Beta, Gamma, Delta, and the most recent Omicron variant of coronavirus [2,3]. This makes it difficult for medical researchers to develop a proper treatment and remedy plan for infected people. Consequently, prevention, early diagnosis, and understanding of the viral phenomena are the keys to saving human lives from these life-threatening diseases.

For diseases like COVID-19 or pneumonia, in general, the infected lung becomes inflamed and may become filled with fluid. Once infected, a patient may experience many signs and symptoms such as fever, cold, cough, and headache. Besides looking for these indicators, medical practitioners usually perform some assay for the accurate diagnosis of the diseases. For example, the current COVID-19 examination involves collecting nose and throat swabs from patients and performing the polymerase chain reaction (PCR) test to detect the presence of viral RNA or using rapid antigen tests to detect viral protein fragments in suspected patients [4]. Medical imaging, such as chest X-ray (CXR) or Computational Tomography (CT) scan, is another common and convenient way to diagnose a lung infection disease [5,6,7]. The CT scan or CXR films of infected people appear with some vascular markings and hazy lung regions, commonly known as ground-glass opacity (GGO) [8]. Medical practitioners need to carefully assess the GGO for proper disease diagnosis, which is time-consuming and sometimes tiresome. The use of CXRs for diagnosis requires a high level of skill, experience, and concentration from the radiologist. Millions of CXRs are generated annually, and a radiologist has to examine a voluminous number of CXR films, culminating in an infeasible diagnostic workload [9]. Moreover, the COVID-19 pandemic has elevated the number of chest radiographs in clinical laboratories, resulting in increased pressure on radiologists and healthcare providers. Consequently, according to the Association of American Medical Colleges, it is estimated that the United States could experience a shortfall of radiologists by 2033 [10].

Computer-aided diagnosis (CAD) systems effectively reduce the diagnostic workload and aid the radiologist or primary doctor in making swift decisions [11]. It plays a supporting and explanatory role in diagnosis using medical imaging. Medical imaging deals with the information in an image that doctors use to evaluate and analyze abnormalities quickly. Analysis of images in the medical field using the CAD system is crucial because it requires a high level of sophistication and accuracy. Researchers nowadays are putting their efforts into making the CAD system more intelligent, accurate, and robust with the help of artificial intelligence (AI), computer vision, and machine learning (ML) techniques [12,13,14]. The main goal of CAD systems is to identify abnormal signs like GGO and extract useful information like shape, size, and density to serve as an automated decision support system for professionals.

The recent advancements in artificial intelligence, especially convolution neural networks (CNNs), have made the potential of automated medical image analysis nearly comparable to that of health professionals. CNNs are now being used in many applications ranging from industry to healthcare and marine object detection to space exploration [15,16,17]. The application of CNNs shows promising performance in object classification, detection, segmentation, and the extraction of useful information without human involvement. Unlike other application fields, investigation of medical imaging must be both sensitive and not overly produce false-negative cases, especially for highly contagious diseases like COVID-19. A false-negative decision of a COVID-19-positive patient could severely jeopardize the patient and the community. While examining the CXR films of a virus or bacterial-infected patient, doctors visually observe CXR films and make a diagnosis. Consequently, an enhanced visualization with a colored scheme can ease the observation process. However, contemporary techniques used for COVID-19 or pneumonia diagnosis are more focused on the task of disease classification and detection and are monochromatic to provide any visual aid.

In response to this challenge, we propose a machine-learning-enabled diagnosis technique with improved visualization of viral or bacterial infected lesions in CXR, using the multi-layer gradient class activation map (ML-Grad-CAM) techniques. The proposed framework consists of three major steps: (1) segmentation-based region of interest (ROI) cropping and data augmentation; (2) a CNN classifier for normal, COVID-19, and pneumonia CXR classes; and (3) a Grad-CAM algorithm to improve the visualization with a saliency map. We also calculate a severity assessment index (SAI) to provide a measure of severity of the infection, which could be an effective indication for prognosis purposes. Thus, major contributions of this proposed approach read as follows:Improve multi-class classification accuracy by using segmentation-based ROI cropping and transfer learning;Improve visualization of disease lesions in CXR films using lower-level information in the saliency map; we named it the ML-Crad-CAM (multi-layer Grad-CAM) algorithm;Propose severity assessment index (SAI) value from the saliency map as an indication of the severity level, which could be used for prognosis purposes.

The rest of the paper is organized as follows. Section 2 provides a brief overview of the latest related work in this field. In Section 3, we present a detailed description of the dataset and the proposed research framework. In Section 4, we discuss the experimental design and outcomes of the three experiments i.e., whole CXR dataset unbalanced (original dataset), whole CXR dataset balanced, and cropped ROI dataset balanced. In Section 5, we present and compare the visualization of the saliency map using the ML-Grad-CAM algorithm. This section also defines and calculates the Severity Assessment Index (SAI) value. Finally, in Section 6, we conclude with some future research directions.

## 2. Related Works

Ayan and Ünver [18] experimented with the Xception and VGG16 models for pneumonia detection based on transfer learning. The test results showed that the VGG16 network exceeded the Xception network at an accuracy of 0.87% over 0.82%. Leveraging the concept of transfer learning and feature extraction, Chouhan et al. [19] developed five different models and used them as an ensemble to increase the accuracy of pneumonia detection using CXR images. The authors reported a 96.4% accuracy score using their developed ensemble deep learning model. Another ensemble approach was proposed by Sirazitdinov et al. [8], where they used RetinaNet and Mask R-CNN for pneumonia detection and localization in CXR images. Besides the regular classification task, this proposed ensemble approach can indicate the infected region in the form of a rectangular bounding box. Jain et al. [20] performed a thorough investigation with two newly proposed CNN architectures and four popular CNNs, namely VGG16, VGG19, ResNet50, and Inception-v3, to classify CXR images into two classes viz., pneumonia and non-pneumonia. The authors achieved the highest accuracy score of 92.31% for the second proposed CNN model, where they used three convolution layers with the adjustment of necessary hyperparameters. Liu et al. [21] proposed the SysFormer framework for tuberculosis diagnosis to leverage the bilateral symmetry property of CXR images in an attempt to improve diagnosis. The authors investigated the SysFormer approach on the Tuberculosis X-ray (TBX11 K) dataset and observed an accuracy of 95.1%.

As COVID-19 emerged late 2019, many researchers began exploring in the use of deep learning techniques in application to COVID-19 diagnosis. For example, Wang et al. [22] demonstrated convolutional transfer learning (TL) methods using five (VGG16, InceptionV3, ResNet50, DenseNet121, and Xception) pre-trained deep learning models and found the Xception model to be the leading one producing an accuracy of 96.7%. Using a similar approach, Abbas et al. [23] proposed a deep CNN framework, called DeTraC (Decompose, Transfer, and Compose), to classify COVID-19 CXR films. The DeTraC method can classify COVID-19 films with 93.1% accuracy. Another CNN-based deep learning model was proposed by El-Rashidy et al. [24] for COVID-19 detection based on patients’ X-ray scan images and transfer learning. Their proposed model achieved a promising result with an accuracy of 97.9%. Minaee et al. [25] used the transfer learning method to investigate four popular convolutional neural networks, including ResNet18, ResNet50, SqueezeNet, and DenseNet-121, to discriminate COVID-19 X-ray films from normal films and obtained the highest accuracy of 92.3 with the SqueezeNet. Ismael and Şengür [26] introduced a new CNN model by integrating deep feature extraction and Support Vector Machine (SVM) to differentiate COVID-19 and normal (healthy) CXR images. The combination of deep feature extraction and SVM classifier with linear kernel function reported a classification accuracy of 94.7%. Nayak et al. [27] explored the impact of hyperparameters of eight CNN models (AlexNet, VGG-16, GoogleNet, MobileNet-V2, SqueezeNet, ResNet-34, ResNet-50, and Incention-V3) to improve the accuracy of the early COVID-19 screening using CXR images. The study revealed that the ResNet-34 outperforms the other models with an accuracy of 98.33%.

While these methodologies only deal with the binary classification (pneumonia vs. non-pneumonia, TB vs. non-TB, or COVID-19 vs. normal) task, some researchers focused on multi-class (COVID-19, normal, and viral pneumonia) classification tasks. For instance, Chowdhury et al. [28] proposed a parallel-dilated convolutional neural network (CNN) to perform classification tasks among COVID-19, normal, and viral pneumonia films and obtained an accuracy of 96.5%. Similarly, Wang et al. [29] utilized the ResNet-101 and ResNet-152 models with good effects for fusion and dynamically improved their weight ratio during the training process. After training, their model achieved 96.1% classification accuracy on the test data. Another attempt to address the multi-class task was performed by Loey et al. [30]. They proposed a Bayesian optimization based CNN model for the recognition of chest X-ray images using a large-scale and balanced dataset. The proposed CNN model achieved 96% accuracy. Khan et al. proposed a CNN model based on the Xception model architecture for the automatic detection of COVID-19 from CXR images. The proposed model produced a 95% accuracy for a three-class classification task [6]. Rahman et al. [31] utilized four pretrained CNNs to test three schemes of classification task for normal vs. pneumonia; bacterial vs. viral pneumonia; and normal, bacterial, and viral pneumonia with reported accuracy scores of 98%, 95%, and 93.3%, respectively. Hussain et al. [32] proposed a novel CNN model named CoroDet for automatic detection of COVID-19 using CXR and experimented with two-class (COVID-19 and normal), three-class (COVID, Normal, and Non-COVID), and four-class (COVID, normal, non-COVID, and pneumonia) classification problems. The authors reported the accuracy score of 99.1%, 94.2%, and 91.2% for 2-class, 3-class, and 4-class classification problems, respectively.

The research articles available in the literature mostly focused on the task of disease detection or classification using chest X-ray images. Most of them addressed the binary class classification problem i.e., either pneumonia vs. normal or COVID-19 vs. normal. Some researchers have experimented with three-class and four-class classification schemes. It is observed that the accuracy of the binary classification tasks is high, while the multi-class classification problems have comparatively low performance [27,32]. This necessitates the development of more efficacious machine learning models for the multi-class classification task, especially to distinguish among COVID-19, pneumonia, and other common respiratory disease classes. As part of various experiments in this paper, it became apparent that the condition of lungs infected with COVID-19 is very similar to pneumonia infection. Moreover, very few researchers paid attention to enhancing their visualizations in order to ease the task of radiological interpretation. Panwar et al. [33] and Umair et al. [34] deployed the Grad-CAM algorithm to create a class-specific heatmap image overlay to highlight the features extracted in the CXR images. As a matter of fact, the research area of generating class-specific heatmaps for CXR images, and visualization with a colored-based scheme remained under-represented in the literature. The proposed research addresses this research gap by improving the multi-class classification technique while improving the visualization of disease lesions in CXR films using the ML-Grad-CAM (multi-layer Grad CAM) algorithm.

## 3. Methodology

This section describes the methodology of disease diagnosis and the visualization process. The machine learning framework adopted in this study is schematically shown in Figure 1. The framework is divided into two major modules. In the first module, we cropped the region of interest (ROI) from the original CXRs. The second module is designed for classification and visualization using the saliency map. A detailed description of each of the modules is provided in the following subsections.

### 3.1. Dataset

From the onset of the COVID-19 pandemic, the research community has actively engaged with healthcare providers to collect chest X-ray images for the application of machine learning and deep learning on radiology images. In such efforts, teams of researchers [35,36] have collaborated with several universities and individual medical practitioners to create a database of chest X-ray images for COVID-19-positive cases. The images were collected from several sources, such as the PadChest dataset, a German medical school (Figshare dataset), GitHub, Kaggle, Twitter, and the Radiological Society of North America (RSNA). As the images are collected from different sources, the former are not of the same quality. Moreover, the quality of the CXR images may vary due to patient’s health conditions and breathing states, machine setup, and human error. Hence, the quality of the raw images was enhanced using the Gamma correction. See [37] for a more detailed description of gamma correction-based image enhancement. This dataset is publicly available and one of the most widely used datasets for performing experiments related to COVID-19 diagnosis and prediction. This dataset consists of a total of 21,164 CXR images in four categories; of which, the numbers of COVID-19, normal, lung opacity (non-COVID-19 lung infection), and viral pneumonia images are 3616, 10,192, 6012, and 1345, respectively. However, in this work, we considered three categories of CXRs, i.e., COVID-19, normal, and pneumonia. The images are in portable network graphics (PNG) file format with a resolution of 299 × 299 pixels.

### 3.2. ROI Cropping

ROI refers to a meaningful and target region in an image. In this study, our target region is the region within the lung boundary in the CXR images. In medical examinations, the clinicians mainly look into the lung region to identify the presence of any bacterial or viral infections. However, machine learning algorithms, especially CNN models, take the entire CXR images as input and extract computational features for classification and disease prediction. Usually, raw CXR images capture surrounding artifacts, non-lung tissues, and other noises. Due to the presence of extraneous boundaries, the CNN models may learn non-lung features, which may deteriorate the accuracy of diagnosis. To avoid such an issue, the lung region is cropped from the original CXRs before feeding them into the CNN model. To identify the lung region, we rely on supervised lung segmentation techniques. In our earlier work [38], a patch-based and lightweight segmentation network was designed using the U-Net architecture (see Figure 2) for this purpose. One major advantage of this network is that it is lightweight and shows excellent segmentation performance on many datasets. However, the network was trained on the Montgomery County (MC) tuberculosis control program dataset [39] producing an average Dice Coefficient (DC) of 94.21%, whereas this study is based on a different dataset. As the new dataset has different CXR images with four different categories, the pre-trained U-Net is used to extract relevant features based on the principle of transfer learning. Transfer learning effectively allows us to use the weights and biases derived from a pre-trained deep learning model and use it as a starting point for another application. See [40] for a detailed review of transfer learning.

The network is pre-trained to segment individual CXR patches. Patches refer to simply cropped portions from the original images. For any new CXR image, we crop the patches and input them into a pre-trained modified U-Net network. This generates segmented patches as outputs, which are subsequently merged together at the same location as the cropped patches on a 256 × 256 background image. Thus, a whole segmented mask is generated from each CXR image, which is a binary mask of the lung region. Once the lung boundary is segmented, the top left and right bottom coordinates are identified. Using these two coordinates, the ROI is marked in a bounding box (bbox). Notice that, due to the inevitable error of the segmentation network, some portions of the lung region may be erroneously segmented and miss some pixels along the lung boundary. In such cases, the ROIs may miss some useful information along the lung boundary and have a negative effect on the downstream CNN model. To mitigate this risk, a tolerance of 10 pixels is added to the rectangular bounding box, as shown in Figure 3c. In addition, the size of the cropped ROI may differ from image to image, which is further resized to 224 × 224 in the downstream CNN architecture.

### 3.3. Data Augmentation

The performance of machine learning (ML) techniques is greatly affected by an imbalanced dataset. Most of the ML algorithms used for classification are designed with the assumption of an equal number of data for each class. If the dataset is imbalanced, ML algorithms tend to learn features mostly for the majority class and have better predictive performance than for the minority class. In this dataset, we have 3616 COVID-19 images, 6012 pneumonia images, and 10192 normal images with a ratio of approximately 1:2:3. For the training scheme, we took 80% of images from each category, resulting in 2892, 4809, and 8153 training samples for COVID-19, pneumonia, and normal cases, respectively. The dataset is imbalanced as the number of COVID-19 CXRs is one-third of the normal CXRs. Obviously, this could lead to an imbalanced classification problem resulting in poor predictive accuracy for COVID-19 cases. To overcome this issue, data augmentation techniques are employed for the COVID-19 and pneumonia CXR images. All the COVID-19 CXR images are transformed by using the “vertical shift” and “zoom” operations. Thus, the number of COVID-19 CXRs increased 3 times to a total number of 8678 images. Half of the pneumonia images were transformed through either a “vertical shift” or “zoom” operation, which generated a total of 7214 CXRs. We did not perform any augmentation operations on the normal CXRs. After performing the augmentation steps, the dataset became balanced with nearly equal numbers of images for each class. Table 1 shows the number of images before and after augmentation, along with the augmentation techniques for each class.

### 3.4. Classification and Visualization

In this module, we first classified COVID-19, pneumonia, and normal chest films based on the CXR images. Then, the saliency map was used to visualize the suspected infected and suspicious lesion areas in the lung boundary. The details of these two schemes are described next.

Classification: The process of convolution is very effective in capturing the discriminatory features among the different classes of interest. Ideally, low-level convolution layers represent the local texture of the lung infections. As the convolution layer goes deeper, it can extract more abstract but comprehensive features of different types of infections, making the CNN model an outstanding automatic classifier. In the last decade, different types of classifiers have been proposed in the literature. Among them, the VGG16, VGG19, ResNet50, ImageNet, DenseNet, and InceptionV3 are the most popular and widely used classifiers. Compared to these deep learning classifiers, the VGG16 has demonstrated superior performance in many applications for a variety of datasets [41], especially those where lower-level features are paramount to classification. Due to its potential superiority, in this study, the classification scheme is designed based on the VGG16 architecture, as shown in Figure 4.

Unlike the original VGG16 network, we modified the top layers for classifying the three categories of CXR images. Our model consists of five convolution blocks, where each of the first two convolution blocks has two convolution layers and each of the last three convolution blocks has three convolution layers. A 5 × 5 kernel was used for the first convolution block, whereas we set a 3 × 3 kernel for the subsequent blocks. The numbers of filters were set to 64, 128, 256, 512, and 512, respectively. The input image size is 224 × 224 × 3 for both the training and testing sets. After each of the convolution blocks, a maxpooling layer is deployed to downsample the image size by half. Thus, we obtained 7 × 7 × 512 feature vectors at the end of the fifth convolution block, which are then passed through the global average pooling layer. The global average pooling layer takes the average of each of the 512 feature maps and generates a single vector of 512 neurons. One advantage of global average pooling is that it is more inherent to the convolution structure by enforcing correspondences between feature maps and categories [42]. Following the global average pooling layer, the network relates to the output layer through a “softmax” activation function, which creates a probability distribution for the three classes.

Visualization: In deep learning, as convolution layers go deeper, they become more difficult to explain and are considered to be a black box model. In recent years, many researchers have attempted to make deep learning more explainable and sensible. Class activation maps (CAMs) are one such powerful technique that helps researchers understand how the images are being categorized [43]. The primary idea of CAMs is to identify which parts or pixels of any image have a higher contribution to any particular prediction. In other words, CAMs can explain which parts of the image activate the most for any specific class. The activated pixels and parts can be visualized by overlaying a heatmap on the original image. This principle can be applied in conjunction with the above classification scheme, where we design a CNN model to classify the diseases, for example, COVID-19, pneumonia, and normal CXR images in this study. After training the network, a new CXR image is fed into the model to predict the class of the CXR. With the CAM tool, the radiologist will be able to see which parts of the CXR activate the COVID-19/pneumonia/normal class. This is very helpful for two main reasons. First, it can interpret the reasoning behind the classification of CXR images. Inherently, the CNN model can automatically identify the features of the different types of pneumonia lesions from the CXR images, which brings the second benefit of the CAM approach, i.e., it gives the radiologist an idea about the location and severity of the infected lung region.

In this work, we proposed adding the CAM, in conjunction with the CXR classification model, for better visualization. The proposed visualization module is developed based on the Gradient class activation map (Grad-CAM). Grad-CAM uses the gradient information from the last convolutional layer of the CNN to understand each neuron for a decision of interest. To obtain the class discriminative localization map of width *w* and height *h* for any class *c*, the gradient of the score $y_{c}$ is computed with respect to feature maps for a convolution layer. Unlike the Grad-CAM, here, we use the gradient information from multiple pooling layers, i.e., with respect to multiple feature maps instead of the single feature maps of the very last layer of the CNN model, as shown in Figure 5. The primary motivation for using gradient information from multiple layers is to use lower-level but useful information in the saliency map. Using such lower-level information has been proven very effective for the purpose of segmentation and visualization [44]. Multiple class activation maps ($c_{i}$) were determined for each of the pooling layers in the VGG16 network, which were later merged together using distributed weights ($w_{i}$) to obtain the final saliency map using Equation (Equation 1).
(1)$$\mathrm{saliency}\;\mathrm{map}\;\left(s\right)=\frac{1}{\sum_{i=n}^{N}w_{i}}\left(\sum_{i=n}^{N}w_{i}\cdot c_{i}\right),\;\mathrm{where}\;w_{i}=\frac{1}{N-i+1}$$
where N=5 (the number of pooling layers in VGG16 architecture) and *n* is the index of the first pooling layer considered for merging in this model. For example, if we start merging from the very first pooling layers, the value of *n* will be 1. Starting from the second pooling layer will impose the value of 2 for *n*, etc. Further, $w_{i}$ represents the weight for merging a set of pooling layers, which is designed in such a way that it does not invalidate the basic reasoning of the original Grad-CAM algorithm in the literature. For example, in the event of generating the saliency map only from the very last activation map $\left(c_{\left(i=5\right)}\right)$ of the model architecture, Equation (Equation 1) turns out to be as simple as $c_{i}$ since the $w_{i}$ turns out to be 1. However, in the event of using multiple activation maps, we add the weights according to the index of the activation maps. The architecture in Figure 5 generates five activation maps (N=5). The weights are determined based on the information intensity and priority of the pooling layers. Obviously, the last pooling layer comprises more comprehensive features and useful information compared to the other pooling layers. Hence, we set the larger weight for the very last layer and reduce the weight for the previous layers according to Equation (Equation 1), depending on the number of merging layers. For example, the third activation map (with *n* = 3) will have a weight of 1/(5−3+1)=0.33. It is observed that merging all five pooling layers sometimes deteriorates the visualization by introducing some surrounding noises into the saliency map. On the other hand, using only the very last pooling layers sometimes misses some useful areas to highlight. Based on our experiment, in this work, we use the last three pooling layers to visualize the saliency map. The detailed results and explanation are reported in Section 5.

## 4. Experimental Design and Results

As illustrated in Section 3.2, before applying augmentation techniques, we split the images into three sets, 80%, 10%, and 10%, for training, validation, and testing, respectively, for each category. This maintains the proportion and inclusion of each category of images into each set of training, testing, and validation data. Thus, we have 18,854 images for training, 1982 images for validation, and 1984 for testing, as reported in Table 2. To train the model, we used a well-tuned set of hyperparameters to ensure optimal performance and generalization. The Adam optimizer [45] was employed with a learning rate of 0.0004, facilitating efficient learning with adaptive adjustments. A batch size of 16 images was used to balance the memory usage and training stability. The “He initialization” [46] was used to ensure proper scaling of weights at the start. To mitigate overfitting, we included a weight decay ($\ell_{2}$ regularization) of $10^{-4}$ and a dropout rate of 0.2 [47]. The model was trained for 50 epochs with early stopping criteria of 10 epochs to halt training if the validation performance plateaued. The loss is calculated by using the categorical cross-entropy [48]. The reason for choosing the categorical cross-entropy is the one-hot encoding format of the label, which makes our results comparable with other existing works in the literature that also used the same loss function (Table 3). The model produces the final maximum training accuracy of 0.9855 and validation accuracy of 0.9622 at the 48th epoch. The history of model training is demonstrated in Figure 6. All the training of CNNs is implemented using TensorFlow on a standard PC with a single 16 GB memory GPU GeForce GTX 1080 Ti. The entire training takes about 12 h. We also used the weights of the trained model to calculate the gradient with respect to each of the feature layers and find the saliency map.

As mentioned earlier in Section 3.1 and Section 3.2, we cropped the ROI and balanced the dataset to improve the training accuracy. To justify the adaptation of these two steps, we conducted three experiments: (1) Exp-1: train and observe the accuracy of the classification model using whole CXR images unbalanced; (2) Exp-2: train and observe the accuracy of the classification model using whole CXR images balanced (without cropping ROIs); and (3) Exp-3: train and observe the accuracy of the classification model using ROI CXR images balanced. Figure 6 demonstrates the training history for experiment 3 only. After the completion of training, each experimental model was tested using unseen test data. The confusion matrix of the results is shown in Figure 7.

Ablation Study: The purpose of this ablation study is to investigate the performance of the trained model in different experimental settings by removing certain steps to understand the contribution of the model to the overall performance. In Figure 7, we observed that the model produces many wrong predictions when the unbalanced and whole CXR images are used for training the classification network (Exp-1). However, balancing the dataset improves the prediction accuracy for all three cases (Exp-2). As expected, cropping the ROIs further improves the prediction results. In comparison to the pneumonia cases, the model was underserved for the COVID-19 and normal cases. It predicts many COVID-19 cases as normal films and vice versa. This confusion arises due to the lack of distinguishable phenomena between COVID-19 and normal CXR films, especially at the early stage of COVID-19 infection. This convoluted situation is greatly improved when we train the model using the balanced and ROI images (Exp-3). In addition to the confusion matrix, the performance of these experiments is measured using accuracy, precision, and recall as per Equations (2)–(4):(2)$$Accuracy_{Overall}\left(Acc\right)=\frac{\mathrm{Number}\;\mathrm{of}\;\mathrm{correct}\;\mathrm{prediction}}{\mathrm{Total}\;\mathrm{number}\;\mathrm{of}\;\mathrm{prediction}}$$
(3)$$Precision_{i}\left(P\right)=\frac{\mathrm{True}\;positive_{i}}{\mathrm{True}\;positive_{i}+\mathrm{False}\;positive_{i}}$$
(4)$$Recall_{i}\left(R\right)=\frac{\mathrm{True}\;positive_{i}}{\mathrm{True}\;positive_{i}+\mathrm{Flase}\;negative_{i}}$$
where *i* represents the class (COVID, pneumonia, and normal) of the prediction. These three metrics are widely used to gauge the performance of deep learning models for classification tasks. Notice that we measure the overall accuracy of the predicted model; however, the precision and recall are measured by class as this is a multiclass classification task. The metrics are calculated based on the test dataset and are documented in Table 4. Clearly, balancing the dataset with ROI cropping significantly increases the performance of the trained model.

A comparative study has been performed with the existing literature to assess the performance of our proposed methodology. To maintain the consistency of comparison, we only compare our results with the approach that used the dataset, as described in Section 3.1. The comparative results are documented in Table 3.

## 5. Visualization and Discussion

Based on the above experiments and observation, we train the deep learning network using the ROI CXR images to build our final classification models. Once trained, the model is used in combination with the multi-layer Grad-CAM (ML-Grad-CAM) algorithm to generate the saliency map for each of the test images. Later, the saliency map is overlaid on the original to visualize the heatmap for predicting the three different categories of CXR images. We also compare the heatmap generated using the ML-Grad-CAM with the original Grad-CAM architecture. The comparison of different heatmaps is depicted for the COVID-19, normal, and pneumonia cases in Figure 8, Figure 9, and Figure 10, respectively.

The heatmap provides a visual cue for the potential infectious region in the CXR films, leading to quick and easy diagnosis. The red dotted circles indicate the infectious region in the original CXRs images, which are clearly highlighted in the heatmap images. Additionally, it can serve the purpose of identifying the severity of the infections by virtue of the heatmap’s spread. Usually, a heatmap highlights the regions (image pixels) with the most contribution for a particular class or category of interest, which, in other words, is the class activation map. For example, in predicting positive and negative classes, the machine learning model must learn to identify which pixels are contributing toward the positive class and negative class. The Grad-CAM algorithm can take the gradient for any particular class and highlight the features (pixels) in the heatmap. Thus, CXR films with fewer infection symptoms, alternatively considered as images with fewer features, should have fewer pixels to be highlighted in the Grad-CAM heatmap, as shown in Figure 9 (images with normal CXR films). On the other hand, as shown in Figure 8 and Figure 10, the CXR film indicating high viral/bacterial infections drives the generation of an intensive heatmap as compared to the normal case or descent CXR images.

From this perspective, the heatmap could be used to interpret the severity of bacterial infection in the lung region. To this end, we calculated a severity assessment index (SAI) for each of the generated saliency maps. The severity assessment index is derived from the intensity of each pixel in the saliency map and expressed in a single numeric value as the indication of the overall severity (the higher the value, the more severe the infection). We take the average over all the numerical values in the saliency map for a given image and use the resulting score as an overall index of the infection in the lung region. Thus, we define the severing assessment index (SAI) as Equation (Equation 5):(5)$$\mathrm{SAI}=\frac{1}{7\times 7}\left(\sum_{i,j}^{M,N}s_{i,j}\right)$$
where, $s_{\left(i,j\right)}$ are the intensity of each pixel of the saliency map (S) and 7×7 is the dimension of the matrix *S*. We evaluate the SAI values for the three categories of CXRs: COVID-19, normal, and pneumonia. The distributions of the SAI scores for each category are shown in Figure 11. Clearly, most of the CXRs of each category contain low to medium levels of infections. A comparably low number of CXRs have a high level of severity. The SAI score is a useful indication of the severity level of infection in the lung region. To make it obvious, the SAI scores for some selected CXR images are shown in Figure 12, along with the corresponding heatmap.

As can be seen, images that present a higher infection (COVID-19 or pneumonia) have higher SAI values, while the images with normal cases have comparatively low values. For example, the first CXR film of row 1 in Figure 12, having fewer lesions in the lung, generates a low value of SAI (0.296). However, images with increased lesions (third image of row 1) generate an intensive heatmap and a higher SAI value of 0.475. We observe the same trends as the CXR examples in the pneumonia class. As reported in the third row in Figure 12, we obtain a low SAI value of 0.322 and a high SAI value of 0.419, depending on the lesion in the lung region. For the CXRs of the normal class, the generated heatmaps are generated with very minimal pixel intensities and, thus, provide low values of SAI close to zero, indicating a healthy lung.

## 6. Conclusions

In this paper, we propose a deep learning technique for the diagnosis of COVID-19, pneumonia, and normal (healthy lung) cases using CXR images with an improved visualization of the saliency map. To improve the accuracy of COVID-19 diagnosis, we experiment and train a modified VGG-16 network with raw CXR films, segmentation-based cropped lung region (ROI), and data balancing. As expected, the model trained with cropped ROI and the balanced dataset performs better than other experimental setups and attains an overall accuracy score of 96.44%. We also propose the multi-layer Grad-CAM algorithm to generate a better saliency map and improve the visualization of suspicious regions in the lung region. The saliency maps generated from the ML-Grad-GAM algorithm are compared with the original Grad-CAM algorithm. Visualization with improved saliency maps can help medical practitioners locate the infected location or identify the presence of ground glass opacity (GGO) in CXR films. In addition, the severity assessment index (SAI) value calculated from the saliency map provides a quantitative measure to understand the level of severity of infection, for any given patient. Such a machine-learning-enabled diagnostic system could be an effective tool for medical practitioners when they need to examine a high number of CXRs each day, especially in situations similar to COVID-19 pandemic. It should be acknowledged that this diagnostics system must not fully replace doctors or radiologists, but it could serve as a decision support system by reducing the overall workload and patients’ waiting time by making the diagnostic process quick and easy. In this study, we deal with classifying three categories of CXRs using publicly available datasets. Our future effort will include extending our model for clinical COVID-19 and other viral/bacterial infection cases and generalizing the model with a more sophisticated deep learning algorithm to make this clinically applicable.

## Figures and Tables

**Figure 1 diagnostics-14-01699-f001:**
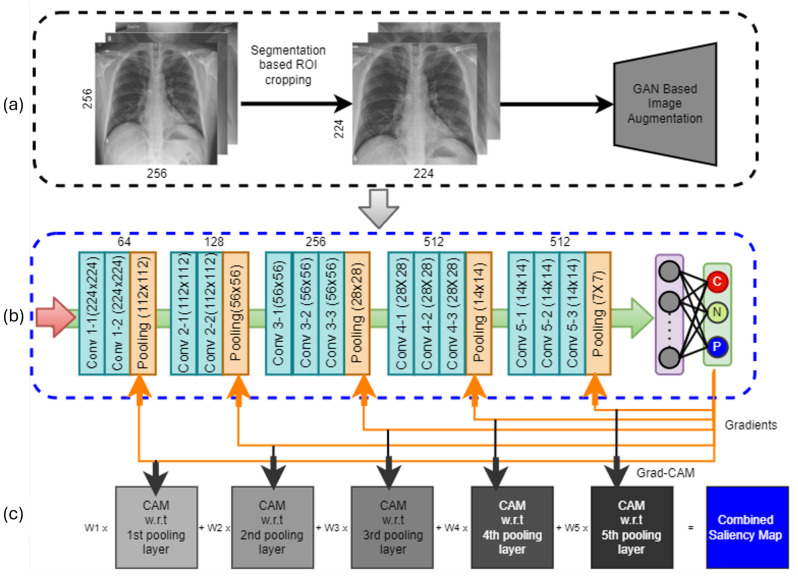
The framework of CXR image classification and lesion visualization; (**a**) image preprocessing with ROI cropping and augmentation, (**b**) multi-class classification with modified VGG16 architecture, and (**c**) the integration of the ML-GRAD-CAM algorithm for visualization.

**Figure 2 diagnostics-14-01699-f002:**
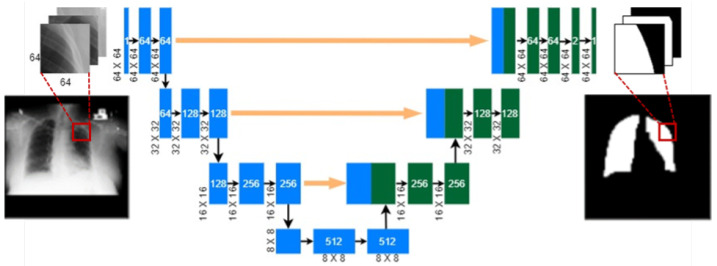
A patch-based U-Net segmentation network [38] was used for transfer learning. Cropped patches were fed into the network to segment individual patches and later merged together to obtain the whole lung region segmentation.

**Figure 3 diagnostics-14-01699-f003:**
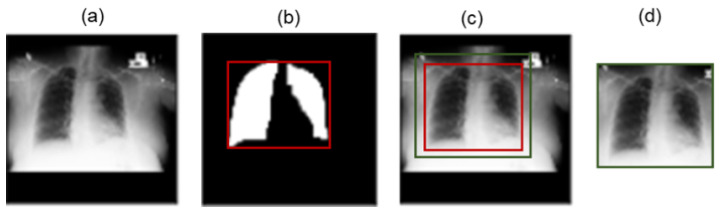
Cropping ROI from CXR: (**a**) original CXR image, (**b**) segmented mask with tight rectangular bbox, (**c**) bounding box around the lung region with additional 10 pixels (green), and (**d**) cropped ROI.

**Figure 4 diagnostics-14-01699-f004:**
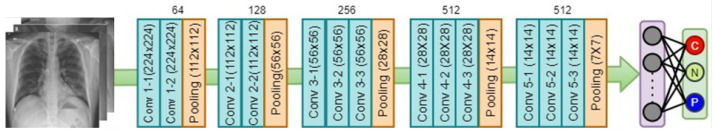
The CNN model for CXR image classification.

**Figure 5 diagnostics-14-01699-f005:**
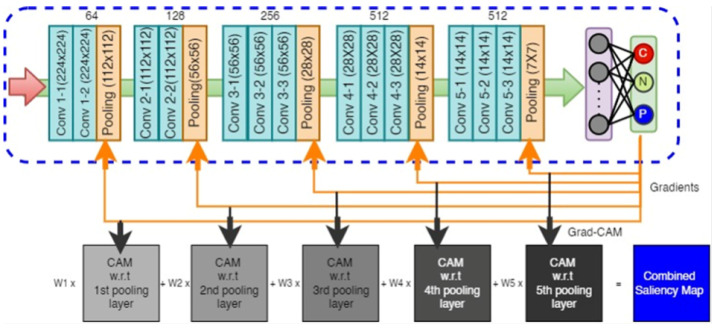
Architecture of multi-layer Grad-CAM (ML Grad-CAM).

**Figure 6 diagnostics-14-01699-f006:**
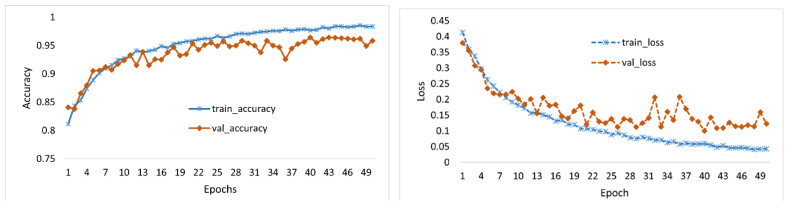
History of the model training by epoch (**left**) training and validation accuracy and (**right**) training and validation loss.

**Figure 7 diagnostics-14-01699-f007:**
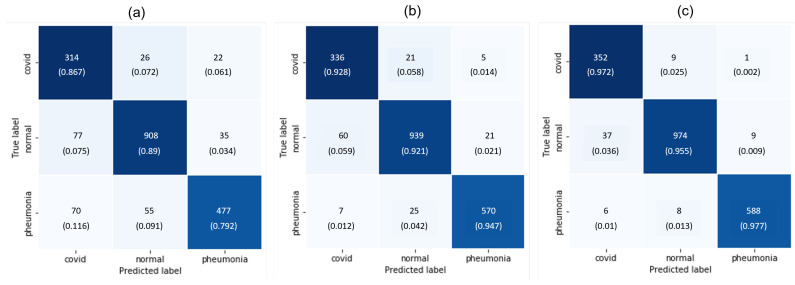
Confusion matrix: (**a**) using whole CXR images unbalanced; (**b**) using whole CXR images balanced (without ROI cropping); and (**c**) using ROI CXR images balanced.

**Figure 8 diagnostics-14-01699-f008:**
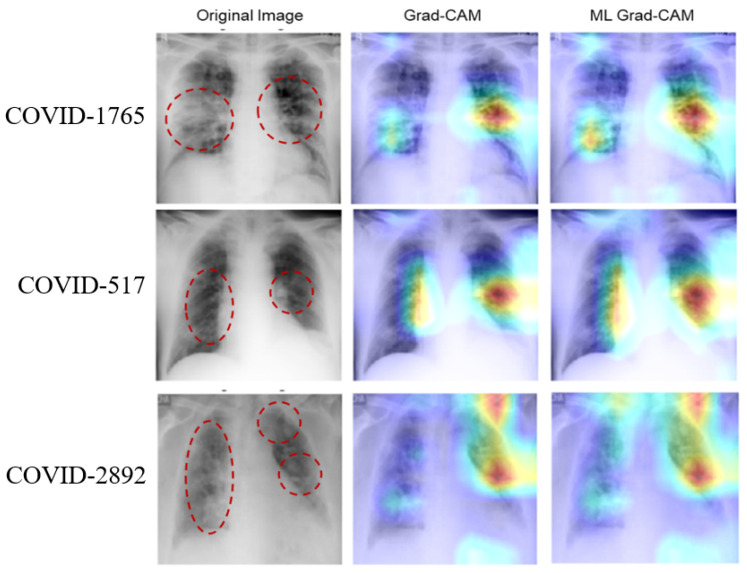
Comparison of heatmaps for COVID-19 CXR images. The rows show three randomly selected images from the COVID-19 test dataset. Column 1 shows the original images with potential suspected lung regions, column 2 shows the heatmap from the Grad-CAM algorithm, and the last column shows the heatmap obtained from the proposed ML Grad-CAM algorithm.

**Figure 9 diagnostics-14-01699-f009:**
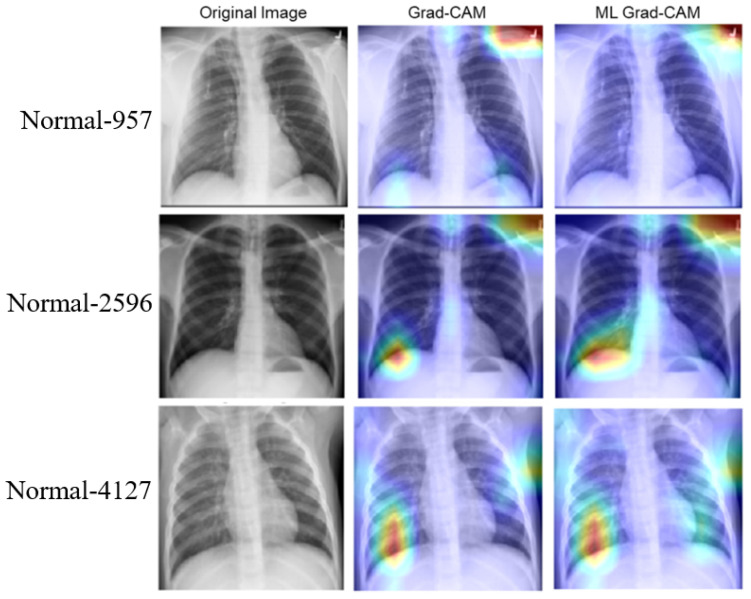
Comparison of heatmaps for normal CXR images. The rows show three randomly selected images from the normal test dataset. Column 1 shows the original images, column 2 shows the heatmap from the Grad-CAM algorithm, and the last column shows the heatmap obtained from the proposed ML Grad-CAM algorithm.

**Figure 10 diagnostics-14-01699-f010:**
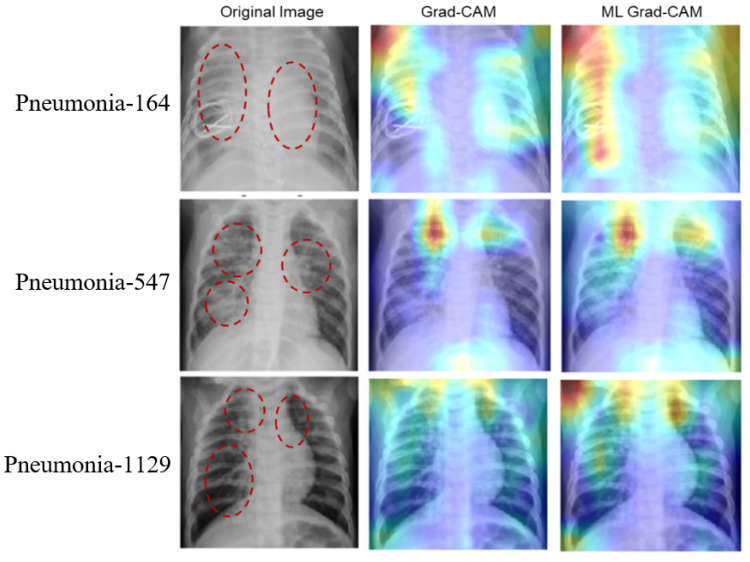
Comparison of heatmaps for pneumonia CXR images. The rows show three randomly selected images from the Pneumonia test dataset. Column 1 shows the original images with potential suspected lung regions, column 2 shows the heatmap from the Grad-CAM algorithm, and the last column shows the heatmap obtained from the proposed ML Grad-CAM algorithm.

**Figure 11 diagnostics-14-01699-f011:**
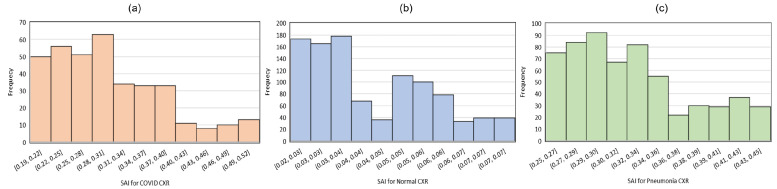
Distribution of SAI Scores for (**a**) COVID-19 CXRs, (**b**) normal CXRs, and (**c**) pneumonia CXRs.

**Figure 12 diagnostics-14-01699-f012:**
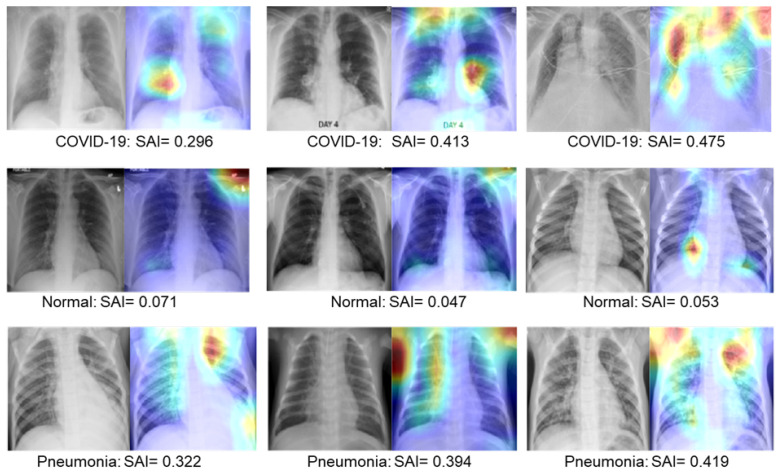
Examples of CXR images with their corresponding heatmaps and SAI values. The first row shows examples of COVID-19 classes with their corresponding heatmap and SAI values. Similarly, the second and third rows show examples of the normal and pneumonia classes, respectively.

**Table 1 diagnostics-14-01699-t001:** The number of CXR Training Samples before and after data augmentation.

Class	Before Augmentation	Augmentation Techniques	After Augmentation
COVID-19	2892	“Vertical shift” and “Zoom”	8678
Pneumonia	4809	“Vertical shift” or “Zoom”	7214
Normal	8153	“None”	8153

**Table 2 diagnostics-14-01699-t002:** Data split for experimental design.

Class	Training	Validation (10% of Original Data)	Test (10% of Original Data)
COVID-19	2892	362	362
Normal	8153	1019	1020
Pneumonia	4809	601	602
Total	15,854	1982	1984

**Table 3 diagnostics-14-01699-t003:** Comparative study of the performance in terms of accuracy.

Reference	Methid	Classification Task	COVID	Pneumonia	Noraml	Accuracy
[22]	TL	Binary	565	-	537	96.7%
[23]	TL	Binary	105	-	80	93.1%
[24]	DL	Binary	250	-	500	**97.9%**
[25]	TL	Binary	184	-	5000	92.3%
[28]	CNN	Multi-Class	219	1345	1341	96.5%
[29]	ResNet	Multi-Class	140	9576	8851	96.1%
[30]	DL	Multi-Class	3616	3616	3616	96.0%
**Proposed Method **	DL	Multi-Class	362	602	1020	**96.5%**

**Table 4 diagnostics-14-01699-t004:** Performance measures of the different experiments on the test dataset.

Experiments	Overall Acc.	P(C) *	R(C)	P(P)	R(P)	P(N)	R(N)
Exp-1	0.856	0.681	0.867	0.893	0.792	0.918	0.890
Exp-2	0.930	0.834	0.928	0.956	0.947	0.953	0.921
Exp-3	0.965	0.891	0.972	0.983	0.977	0.983	0.955

* C, P, and N represent the COVID, pneumonia, and normal classes, respectively.

## Data Availability

The dataset is publicly available at https://www.kaggle.com/datasets/tawsifurrahman/covid19-radiography-database accessed on 22 March 2022, and the source code will be available upon request.

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
