# Peer review of "Machine-Learning-Enabled Diagnostics with Improved Visualization of Disease Lesions in Chest X-ray Images"

_diagnostics, 2024, doi:10.3390/diagnostics14161699_

Round 1
Reviewer 1 Report
Comments and Suggestions for Authors
The present manuscript entitled “Title: Machine Learning Enabled Diagnostics With Improved Visualization of Disease Lesion in Chest X-Ray Images” introduces a deep learning approach based on VGG-16 architecture, enhanced by the ML-Grad-CAM algorithm, to improve the accuracy and visualization of diagnosing COVID-19 and pneumonia in chest X-ray images. The proposed model achieves an accuracy of 96.44% in classifying CXR images into three categories: COVID-19, normal, and pneumonia. This study utilizes a dataset of 21,164 CXR images. Focuses on lung regions to enhance model accuracy. Improves data balance through augmentation techniques. Introduces the Severity Assessment Index (SAI) as a quantitative measure of infection severity.
The methodology presented in the manuscript is sound, employing appropriate preprocessing techniques and data augmentation to achieve a balanced dataset for classification. The application of zoom and vertical shifts is particularly effective. The number of objects after augmentation across all three groups is adequate. The concept of an interpretable network utilizing the Grad-CAM algorithm is commendable.
However, a few points require attention:
1. In Figure 1, part (a), it is essential to indicate the initial size of the images and the dimensions they reach after cropping. Please specify the final size of the cropped images.
2. Regarding the loss function, while categorical cross-entropy was utilized, it would be beneficial to explain why the sparse method was not chosen, as it often yields favorable results for multi-class classification.
3. In Table 4, highlighting the best results in bold would enhance clarity for readers.
Reviewer 2 Report
Comments and Suggestions for Authors
The authors present improved detection, classification and visualization of disease lesions in chest X-ray images. They modify VGG-16 in combination with image enhancement, segmentation of region of interest, and data augmentation that improve classification accuracy. Meanwhile, multi-layer Grad-CAM algorithm is proposed to generate class-specific saliency map that improve visualization of suspicious regions. Later, a severity assessment index is defined from saliency map for measuring infection severity. Interestingly, the severity assessment index adds additional meaning to the visualization and provides good interpretation of disease severity. The work is well organized, read with interest, and easy to follow. It could be accepted in the current form.
Reviewer 3 Report
Comments and Suggestions for Authors
This paper describes a CNN model with visualization for diagnosing chest X-ray Images. The followings are some comments:
1. Section 3.1 discusses the used dataset. Does it have a name? Any link to the dataset?
2. Is the image size after segmentation a constant? If not, how to handle variable image sizes in the proposed CNN?
3. Data augmentation should be applied exclusively to training samples. Using an augmented dataset to compute testing accuracy can bias the results. Additionally, there is a risk of data leakage if two augmented samples from the same source are used, one in training and the other in testing.
4. Eq. 1 provides a formula to compute the weights for each activation map. I understand that maps with larger i is usually more important. Still, I would like to see a discussion about the design of the weights.
5. Please use percentage in Fig. 7. It is difficult to compare the relative performance in the present form. By the way, I suppose that Fig. 7(a) is the results for testing the original (not augmented) dataset. In that case, the test set should have fewer samples than the other two. But, the figure shows otherwise. Please explain.
6. Section 5 shows the effectiveness of the SAI with some examples. It would be more convincing if the authors can provide three histograms of the SAI, one for each class. Showing histograms can better examine the effectiveness.
Reviewer 4 Report
Comments and Suggestions for Authors
The manuscript proposes improving classification accuracy using a VGG-16 architecture-based deep learning approach combined with image enhancement, segmentation-based ROI cropping, and data augmentation. It outlines a comprehensive methodology, including the use of ML-Grad-CAM for improved visualization and the development of a Severity Assessment Index (SAI) to measure infection severity.
There are several points that the authors are required to address
· The abstract contains technical terms and acronyms (e.g., VGG-16, Grad-CAM, ROI, SAI, CXR) that might not be immediately clear to all readers. Brief explanations or expansions of these terms could improve accessibility
· While the abstract mentions improvements in classification accuracy and visualization, it does not provide specific details on how these enhancements compare quantitatively to existing methods.
· The employed dataset used for training and testing is not directly stated to provide context and support the validity of the results.
· In fig 6 there are variation right part which might indicate that the learning rate is set too high
· The method hyperparameters are not given which makes the replicability of the method is limited
Comments on the Quality of English Languageminor English editing is required
Author Response
Please see the attachment."

Round 2
Reviewer 3 Report
Comments and Suggestions for Authors
Thanks for the revision.